# Multi-clonal SARS-CoV-2 neutralization by antibodies isolated from severe COVID-19 convalescent donors

Michael Mor[1], Michal Werbner[2], Joel Alter[2], Modi Safra[3], Elad Chomsky[4], Jamie C. Lee[5], Smadar Hada-Neeman[6], Ksenia Polonsky[1], Cameron J. Nowell[7], Alex E. Clark[5], Anna Roitburd-Berman[6], Noam Ben-Shalom[1], Michal Navon[1], Dor Rafael[1], Hila Sharim[4], Evgeny Kiner[4], Eric R. Griffis[8], Jonathan M. Gershoni[6], Oren Kobiler[1], Sandra Lawrynowicz Leibel[5], Oren Zimhony[9], Aaron F. Carlin[5], Gur Yaari[3], Moshe Dessau[2], Meital Gal-Tanamy[2], David Hagin[10], Ben A. Croker[5], Natalia T. Freund[1] *

1 Sackler Faculty of Medicine, Tel Aviv University, Tel-Aviv, Israel, 2 Azrieli Faculty of Medicine, Bar Ilan University, Safed, Israel, 3 Alexander Kofkin Faculty of Engineering, Bar Ilan University, Ramat Gan, Israel, 4 ImmunAi, New York, New York, United States of America, 5 School of Medicine, University of California San Diego, La Jolla, California, United States of America, 6 George S. Wise Life sciences Faculty, Tel Aviv University, Tel-Aviv, Israel, 7 Monash Institute of Pharmaceutical Sciences, Parkville, Victoria, Australia, 8 Nikon Imaging Center, University of California San Diego, California, United States of America, 9 Kaplan Medical Center, Rehovot, Israel, 10 Ichilov Hospital, Tel-Aviv, Israel

* nfreund@tauex.tau.ac.il

**Data Availability Statement:** All relevant data are within the manuscript and its Supporting Information files. The sequences related to Fig 2 were uploaded to SRA (Sequence Read Archive)

## Abstract

The interactions between antibodies, SARS-CoV-2 and immune cells contribute to the pathogenesis of COVID-19 and protective immunity. To understand the differences between antibody responses in mild *versus* severe cases of COVID-19, we analyzed the B cell responses in patients 1.5 months post SARS-CoV-2 infection. Severe, and not mild, infection correlated with high titers of IgG against Spike receptor binding domain (RBD) that were capable of ACE2:RBD inhibition. B cell receptor (BCR) sequencing revealed that VH3-53 was enriched during severe infection. Of the 22 antibodies cloned from two severe donors, six exhibited potent neutralization against authentic SARS-CoV-2, and inhibited syncytia formation. Using peptide libraries, competition ELISA and mutagenesis of RBD, we mapped the epitopes of the neutralizing antibodies (nAbs) to three different sites on the Spike. Finally, we used combinations of nAbs targeting different immune-sites to efficiently block SARS-CoV-2 infection. Analysis of 49 healthy BCR repertoires revealed that the nAbs germline VHJH precursors comprise up to 2.7% of all VHJHs. We demonstrate that severe COVID-19 is associated with unique BCR signatures and multi-clonal neutralizing responses that are relatively frequent in the population. Moreover, our data support the use of combination antibody therapy to prevent and treat COVID-19.

with the submission number SUB8326459. BCR sequence data shown in Fig 2 and S4 Fig. The data can be accessed via the following link: https://www.ncbi.nlm.nih.gov/bioproject/PRJNA670581. The neutralising antibody sequences are available from the NCBI GenBank database (accession numbers: TAU-1109_HC_VDJ MW367317 TAU-1109_LC_VJ MW367318 TAU-2189_HC_VDJ MW367319 TAU-2189_LC_VJ MW367320 TAU-1145_HC_VDJ MW367321 TAU-1145_LC_VJ MW367322 TAU-2220_HC_VDJ MW367323 TAU-2220_LC_VJ MW367324 TAU-2230_HC_VDJ MW367325 TAU-2230_LC_VJ MW367326 TAU-2212_HC_VDJ MW367327 TAU-2212_LC_VJ MW367328 TAU-2303_HC_VDJ MW367329 TAU-2303_LC_VJ MW367330 TAU-2310_HC_VDJ MW367331 TAU-2310_LC_VJ MW367332 TAU-1115_HC_VDJ MW367333 TAU-1115_LC_VJ MW367334).

**Funding:** This research was funded by the Israeli Science Foundation grant 3711/20 (N. T. F. and M. G. T) and by the Tel Aviv University Vice President of Research and Development. N.T.F is also funded by the ISF grant number 41222/18, and Israeli Innovation Authority grant number 68972. This work was also supported by NIH Grant RO1 HL124209 (B.C.), the American Asthma Foundation (B.C.), and the BSF 2017176 (B.C.), a Career Award for Medical Scientists from the Burroughs Welcome Fund (A.F.C.), ISF grants 2475/19, to M. G. T., and 401/18 to M. D., and The Dangoor Centre For Personalized Medicine, BIU (to M. G.T. and M. D.). The funders had no role in study design, data collection and analysis, decision to publish, or preparation of the manuscript.

**Competing interests:** The authors have declared that no competing interests exist The antibodies described in this manuscript are protected by patent and NT Freund and M Mor and D Hagin are the inventors.

## Author summary

The correlates of effective durable antibody response to SARS-CoV-2 infection are still unclear. In this study, we compared B cell receptor signatures in 8 Severe *versus* 10 Mild SARS-CoV-2 infected Israeli donors, at 1.5 months post infection using molecular and bioinformatic approaches. We found distinct features between the two groups with higher anti-SARS-CoV-2 receptor binding domain (RBD) plasma IgG titers and increased B cell expansion in donors with severe disease manifestations. We further isolated 22 monoclonal antibodies from these donors, 6 of which were highly potent neutralizing the live virus and inhibited the fusion of infected cells. Using mutagenesis and peptide libraries we mapped the binding sites of the neutralizing antibodies on the RBD of the SARS-CoV-2 Spike. We next demonstrated that combinations of different classes of neutralizing mAbs can completely block the live virus from spreading in culture. Lastly, we performed a bioinformatic search in 49 healthy BCR repertoires identifying precursors for these neutralizing antibodies in the top 30 most common precursors, suggesting that these antibodies can be readily produced by the majority of the uninfected population upon antigenic stimulation.

## Introduction

The COVID-19 pandemic is caused by the newly emerged SARS-Coronavirus-2 (SARS-CoV-2) and has incomparable public health, economic, and social consequences [1,2]. Generation of neutralizing antibodies (nAbs) that specifically target the Receptor Binding Domain (RBD) of the SARS-CoV-2 Spike protein is thought to be essential in controlling SARS-CoV-2 [3–5]. Support of this assumption is the fact that RBD specific nAbs have been detected and isolated from convalescent patients that have recovered from COVID-19 as early as 8 days post diagnosis [6], exhibiting little clonal expansion and low numbers of somatic hyper mutations [7–10]. However, the critical survey of the humoral response to SARS-CoV-2 infection is somewhat confusing. Some of the recoverees tend to have robust and long lasting immunity while others display a waning of their neutralizing antibodies [6,11]. The correlates of effective durable antibody response are still unclear.

Infection with SARS-CoV-2 varies clinically from asymptomatic in some infected individuals to a severe disease associated with morbidity and mortality in others [12,13]. The development of an effective antiviral antibody response may limit viral dissemination, thereby preventing a severe disease. Alternatively, severe disease can be prevented by T cell-mediated immunity or innate effector mechanisms which independently of B cells restrict viral replication and disease progression. To assess the clinical profile associated with effective antibody responses following SARS-CoV-2 infection, we recruited 18 SARS-CoV-2 infected individuals 6 weeks post infection. Eight out of the 18 donors had been hospitalized due to severe COVID-19 (termed "Severe"), while the remaining 10 donors had experienced mild or no symptoms (termed "Mild"). The intention of the present study was to comprehensively analyze the cellular, molecular, and functional aspects of the antibody responses in Severe *versus* Mild SARS-CoV-2 convalescent donors in order to understand the potential roles of antibodies in COVID-19. Our conclusion is that severe, and not mild infection, is correlated with robust SARS-CoV-2 antibody responses. Furthermore, we isolated 6 neutralizing antibodies from two severely affected donors and characterized their binding mechanisms using peptide libraries, computational methods and mutagenesis while examining their anti-viral efficacy alone and in combination against authentic SARS-CoV-2.

## Results

### Severe COVID-19 disease is accompanied by stronger IgG responses compared to mild

To characterize the antibody repertoire in COVID-19 convalescent donors, we examined B cell responses from 18 donors (S1 Table). Based on clinical information, the donors were divided into two groups; 8 donors had experienced severe disease manifestations following SARS-CoV-2 infection with >1 week of high fever (above 38.5˚C), and pneumonia that required hospitalization. These donors were defined as Severe. The remaining 10 donors were largely asymptomatic following SARS-CoV-2 infection, or had reported only mild symptoms such as loss of taste and/or smell, fatigue, or fever lasting 2 for days or less. These donors were defined as Mild. The Spike protein is responsible for viral attachment and entry into the host cells, and hence, is the target of most of the antibody responses [14–16]. Within the Spike protein, the receptor binding domain (RBD) is the minimal subunit that mediates viral entry [4,5] and was reported to be the main target for neutralizing antibodies elicited during infection [7,17–24]. Therefore, we first asked whether the Mild donors had developed higher antibody plasma responses towards the SARS-CoV-2 Spike RBD (produced in house, S1 Fig). Mild donors had significantly less anti-RBD IgG than Severe donors ($p$ = 0.0008, Figs 1A and S2A), while no differences were observed between Mild and Severe groups for anti-RBD IgM and IgA (Figs 1B, 1C and S2B and S2C). Moreover, ELISA revealed that plasma from Mild donors had significantly weaker ability to inhibit RBD binding to a soluble angiotensin converting enzyme 2 (ACE2, produced in house, S1 Fig) compared to plasma from the Severe donors' group ($p$ < 0.0001, Figs 1D and S2D). As expected, the levels of anti-RBD IgG, but not IgM or IgA, were directly correlated with the ability of the plasma to inhibit RBD:ACE2 binding ($r^2$ = 0.9018, $p$ < 0.0001, Fig 1E–1G). Antibody responses against related respiratory viruses were previously reported to improve the outcome of a SARS-CoV-2 infection [25]. We, however, did not observe differences between the Mild and Severe groups in their IgG titers against the closely related seasonal beta Coronaviruses OC43 and 229E, as well as in IgG titers against Cytomegalovirus (CMV) and Herpes Simplex Virus 1 (HSV1) (S3 Fig). We therefore conclude that, compared to Severe donors, Mild donors have lower titers of SARS-CoV-2 IgG, and a decreased ability to inhibit RBD:ACE2 binding, while no difference in IgG directed against beta Coronaviruses OC43 and 229E.

### Severe Covid-19 patients exhibit distinct BCR signatures

To compare B cell responses in Mild *versus* Severe donors at the molecular level, we performed total B cell receptor (BCR) sequencing using a single cell 10x platform on seven Mild donors (CoV03, CoV04, CoV05, CoV07 CoV10, CoV11, and CoV14, Fig 2A), and six Severe donors (CoV01, CoV02, CoV09, CoV12 CoV13, and CoV16). Insufficient cell amounts were available for analysis from donors CoV08, CoV17, CoV15, and CoV18, and donor CoV06 was excluded due to clinical reasons. When comparing the BCR dataset from the total 13 COVID-19 donors (Mild and Severe donors) to two COVID-19 unexposed BCR datasets (7 BCR repertoires from mature B cells, and 42 BCR repertoires from naïve B cells), 6 VHJH combinations were over-represented, and 3 VHJH combinations were under-represented in COVID-19 samples (marked in green and red, respectively, Fig 2B). When analyzing the light chains, COVID-19 samples had enriched representation of two VKJK combinations, while five VKJK combinations were under-represented when compared to 42 BCR repertoires from naïve B cells (S4A Fig). Next, we compared the BCR sequences between Mild and Severe COVID-19 donors. Mild COVID-19 donors had significantly lower B cell lineage overlap compared to the B cell

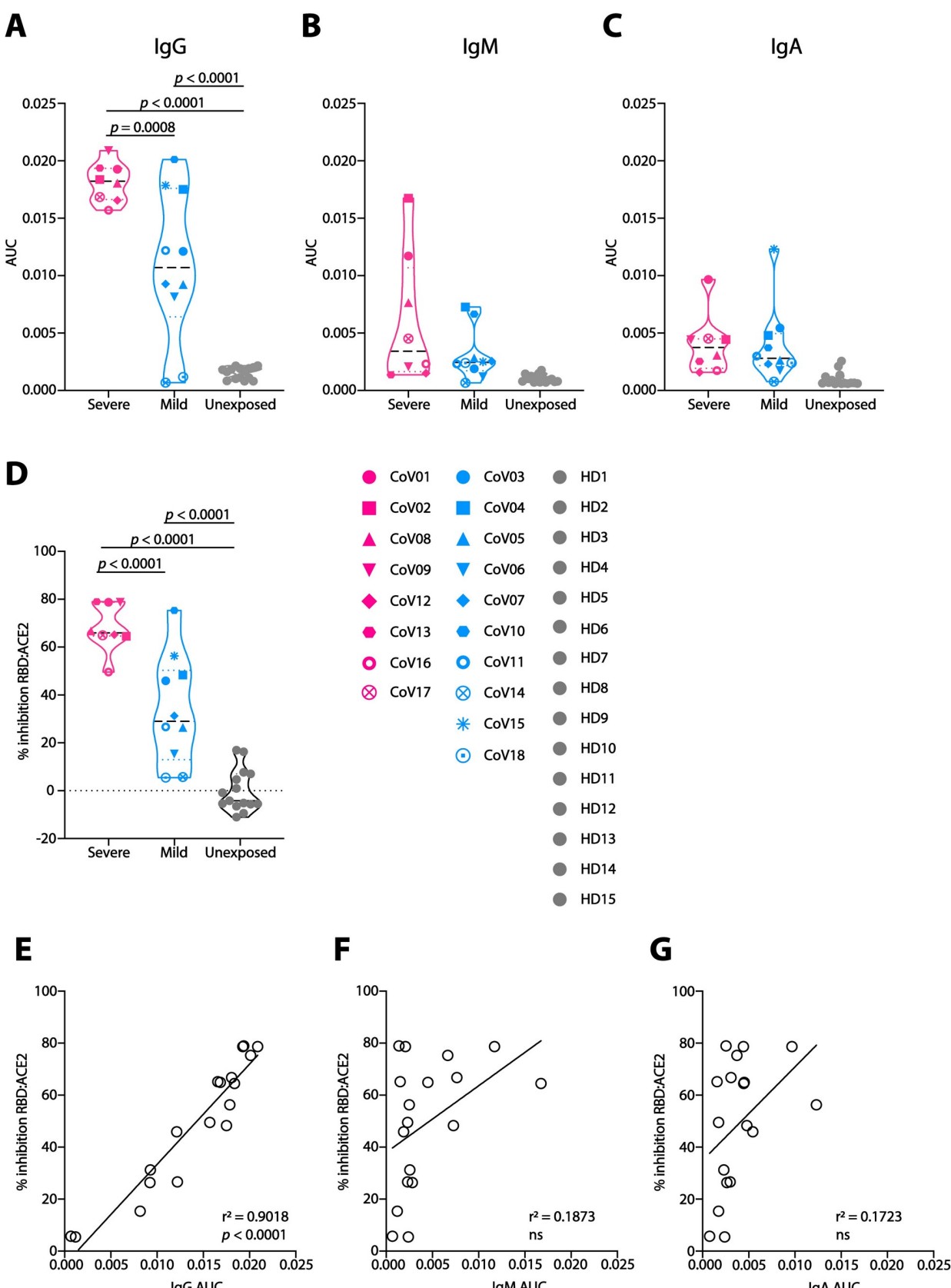

**Fig 1. Plasma responses in COVID-19 convalescent donors.** (**A**)-(**C**) Area under the curve (AUC) values for anti-SARS-CoV-2 RBD IgG (**A**), IgM (**B**) and IgA (**C**). Severe donors are colored in pink, Mild in light blue, and Unexposed in light gray (named "HD1-15", samples collected prior to November 2019). Donors' symbols are indicated on the right side of panel (**D**). (**D**) Inhibition of RBD:ACE2 binding by donors' plasma in ELISA. All samples were normalized to the average value of the healthy donors, that was set as zero inhibition. (**E**)-(**G**) Correlation between plasma inhibition and AUC of IgG (**E**), IgM (**F**) and IgA (**G**). AUC and correlations (simple linear regression) were calculated using GraphPad Prism software. Statistical analysis for panels **A-D** were performed using one-way ANOVA test.

lineage overlap within the Severe group (55 shared B cell clones in Mild, compared to 96 shared B cell clones in Severe, Fig 2C). In addition, VH3-53 gene was significantly over-expressed in the Severe group when compared to Mild ($p = 0.004$, Fig 2D and 2E), whereas no significant differences in VK gene usage was found between the two groups of COVID-19 donors (S4B and S4C Fig). When comparing both heavy and light chains, 18 VHVL combinations were enriched in Severe, and 4 VHVL combinations were enriched in the Mild patients. Overall, these results indicate that severe SARS-CoV-2 infection is associated with unique B cell signatures, enrichment of VH3-53, and a greater B cell clonal expansion.

## 6 mAbs from donors CoV01 and CoV02 effectively neutralize SARS-CoV-2

While the frequencies of anti-RBD B cells were not statistically significantly different between the two groups, on average, Mild donors display lower frequencies of anti-RBD B cells compared to Severe donors (Figs 3B and S5). Taken together with the plasma and the BCR-seq profiling, this suggests that higher B cell responses are associated with severe, and not mild, SARS-CoV-2 infection. We therefore focused on the Severe group of COVID-19 convalescent donors. To investigate antibody responses in the Severe group, we used single B cell sorting method using flow cytometry as previously described [7,26] with SARS-CoV-2 RBD as a bait to derive monoclonal antibodies for analysis. We selected two donors, CoV01 and CoV02, a married couple, who were amongst the first 10 documented COVID-19 cases in Israel (in January 2021, there are aproximately 600,000 positive cases in Israel). CoV01 infected CoV02 following a visit to Italy, as well as two of their three children. Both donors were hospitalized due to severe pneumonia and prolonged high fever. While both donors had strong anti-RBD IgG responses by ELISA (Fig 1A), CoV01 had a lower frequency of anti-RBD B cells compared to CoV02 (Fig 3B). Using single B cell sorting and Sanger sequencing as previously described [26,27] we were able to sequence a total of 35 and 118 RBD-specific paired heavy and light chains from memory B cells of CoV01 and CoV02, respectively (Fig 3C). The two donors exhibited divergent B cell responses. While 19.5% of the sequences extracted from donor CoV02 comprised 8 clonally related families, the sequences of donor CoV01 showed no clonality, possibly due to a lower number of anti-RBD B cells (Fig 3B). The degree of somatic hypermutations were higher in CoV01, while the sequences from CoV02 were similar to germline (Fig 3D). The CDRH3 lengths of the sequences were similar to each other (Fig 3E). Next, we focused on antibodies that were part of the expanded clones as previously described [7,26]. For sequences from donor CoV01, where we did not observe clonal expansion, we selected the most mutated sequences for production. We generated 9 and 13 monoclonal antibodies from donors CoV01 and CoV02, respectively (S2 Table). Eight of these mAbs (3 from CoV01 and 5 from CoV02) bound SARS-CoV-2 RBD (Figs 3F and S6A and S6B left panels and S7) and stabilized Spike trimer [28] protein by ELISA (Figs 3F and S6A and S6B right panels). mAb TAU-1109 exhibited 40% competition with the previously described mAb CR3022 [29], which recognizes a non-ACE2 binding site (non-ACE2bs, Fig 3G) [30]. mAbs TAU-1145, TAU-2189, TAU-2230, and TAU-2303 inhibited RBD binding to both soluble human ACE2 (hACE2) by ELISA (Fig 3H), and to hACE2 expressing cells (Figs 3I and S8).

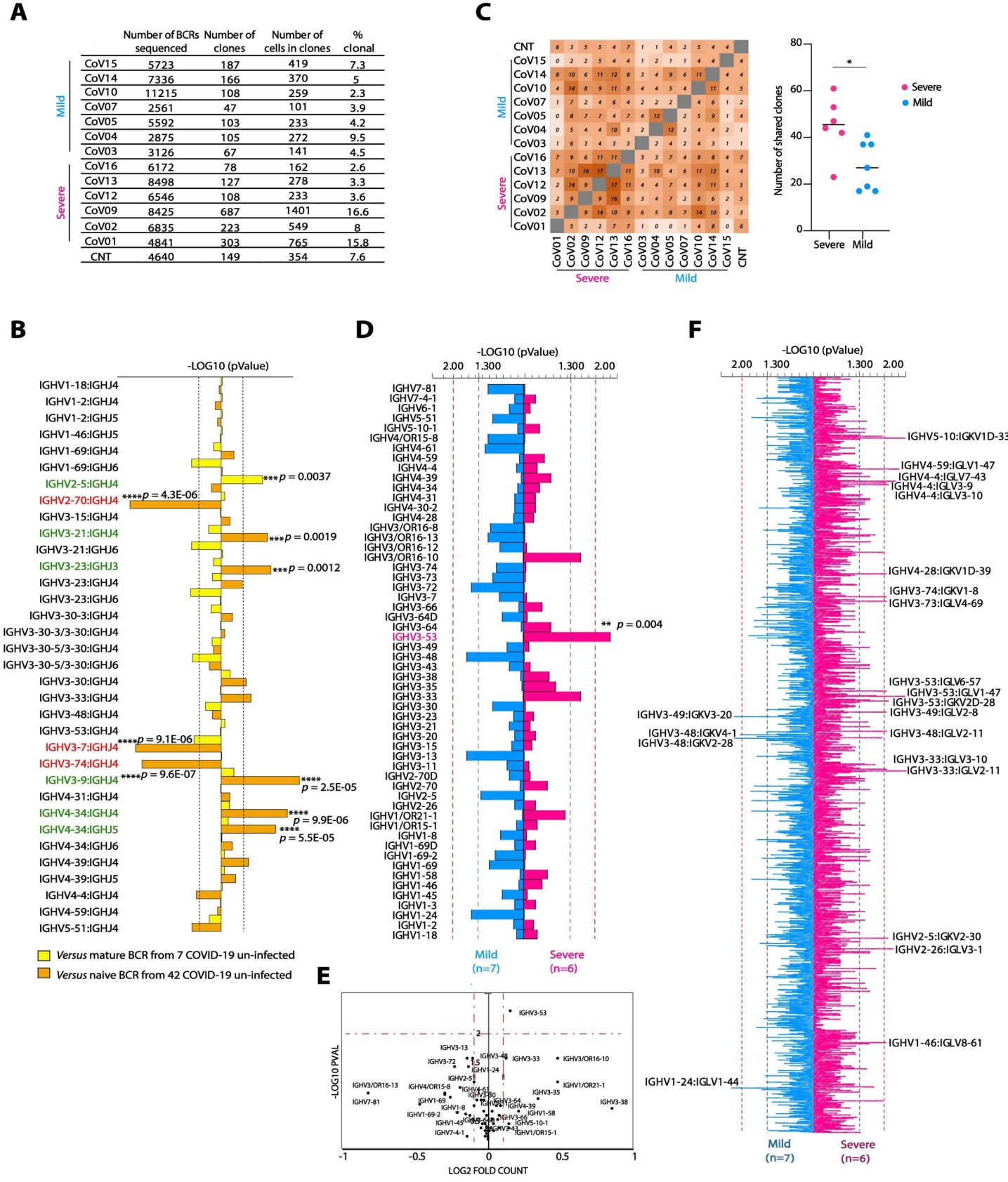

**Fig 2. BCR sequence analysis in convalescent COVID-19 donors.** (**A**) BCR sequencing information for each donor. The total numbers of sequenced BCRs, the number of identified clones (based on similar VHJH and CDRH3), the number of cells in clones, as well as percent of clonal sequences are given for each donor. (**B**) Frequencies of the top 34 VHJH combinations in BCRs from 13 COVID-19 donors *versus* mature BCR from 7 healthy donors (yellow bars) or BCR from naïve B cells from 42 healthy donors (orange bars). -Log$_{10}$ $p$ values were calculated using Mann Whitney test with FDR correction. Over-represented VHJH combinations are marked in green, while under-represented are marked in red. $p$ values are listed by the corresponding horizontal column. (**C**) Left panel: shared clone matrix presented as a heatmap and showing the numbers of clones shared between the different donors. CNT–uninfected control. Right panel: the number of shared clones within the Severe group (magenta) and the Mild group (light blue). Statistical analysis was performed using Student's t test. (**D**) Comparison between the frequencies of 60 VH genes in COVID-19 infected BCR database while comparing between Severe and Mild groups of donors. Magenta, right pointing horizontal bars indicate VH genes that are over-represented in Severe donors over Mild, while light blue, left pointing horizontal bars indicate VH genes that are over-represented in Mild donors over Severe. Blue dashed lines correspond to $p < 0.05$ and red dashed lines corresponds to $p < 0.01$. $p$ values were calculated using Mann Whitney test. The two VH genes that are significantly different between the groups are marked in magenta (**E**) Volcano plot showing $p$ value as calculated in (**D**) and normalized fold counts for every VH segment. (**F**) Comparison between the frequencies of 2500 VH-VL pairs in COVID-19 infected Severe *versus* Mild groups of donors. Only pairs that are below $p < 0.01$ are indicated. Color code, configuration and $p$ values are the same as in (**E**). ** $p < 0.01$ *** $p < 0.001$ **** $p < 0.0001$.

Next, we asked whether the mAbs have neutralizing activity. We first used lentivirus-based GFP-reporter pseudo-viruses that present SARS-CoV-2 Spike on their membrane to infect hACE2-expressing HEK-293 cells (S9 Fig). All the RBD and Spike-binding mAbs had some neutralizing activity in the pseudo-viral assay with IC50 ranging between 0.04 µg/mL to 3.8 µg/mL (Fig 4A). We then tested the neutralizing activity of the mAbs against authentic SARS-CoV-2. Vero E6 cells were infected with authentic SARS-CoV-2 in the presence of mAbs, and after 48 h virus was detected using fluorescently labeled anti-nucleocapsid anti-body. Five of the nine mAbs that neutralized the pseudo-typed virus, also neutralized authentic virus when used at a concentration of 10 µg/mL (TAU-1109, -2212, -2189, -2230, and -2303, Figs 4B, 4C and S10), while four mAbs that demonstrated neutralizing activity in the pseudo-viral assay (TAU-1115, -1145, -2220 and -2310) did not inhibit authentic SARS-CoV-2. To further screen for neutralizing activity of these mAbs, the viability of the cells was tracked by cell imaging for three days to examine the effects of mAb titrations on virus-induced cell death. The same mAbs (TAU-1109, -2212, -2189, -2230, and -2303) dramatically decreased the number of dead cells when used at concentrations ranging from 1–100 µg/mL (Figs 4D and S11), and two of the mAbs, TAU-2212 and TAU-2230 inhibited syncytia formation (Fig 4E and 4F). Interestingly, mAb TAU-2212 demonstrated impressive neutralizing ability in all three assays (pseudo-virus, authentic virus and syncytia formation) despite having no detectable binding to either SARS-CoV-2 RBD or the Spike trimer by ELISA (Fig 3F). Using fluorescent microscopy and flow cytometry, we confirmed the binding of TAU-2212 to the viral Spike when the trimer was transiently expressed on the surface of HEK-293 cells (Figs 4G and S12). It is known that the SARS-CoV-2 Spike protein transitions rapidly between an open and a closed state [31], suggesting that TAU-2212 may be targeting a conformation-dependent epitope, or an epitope that involves more than one subunit.

## G446 is a key residue in the binding of ACE2bs nAbs

We sought to further explore the epitopes of ACE2bs neutralizing mAbs (ACE2bs nAbs) on RBD. We focused on nAb TAU-2230 that exhibited over 80% blockade of ACE2:RBD interaction by ELISA (Fig 3H). To map the epitope of this nAb on the surface of RBD we used phage-displayed random peptide libraries and the Mapitope algorithm [32]. In this method, one maps linear affinity selected peptides pulled from a random peptide library onto the 3-D atomic coordinates of the antigen to get an output of 2–3 epitope candidates [33]. A total of 16 peptides were isolated with TAU-2230 (Fig 5A), and these were used as input for Mapitope to generate an epitope prediction. Three epitope candidates were predicted, Clusters A-C, one of which, Cluster B, overlapped with the ACE2bs (S13 Fig). Cluster B had 4 shared contact residues with ACE2bs [4] (Fig 5B). To identify key residues for TAU-2230 recognition, we introduced a panel of point mutations in RBD in the TAU-2230 predicted epitope, as well as

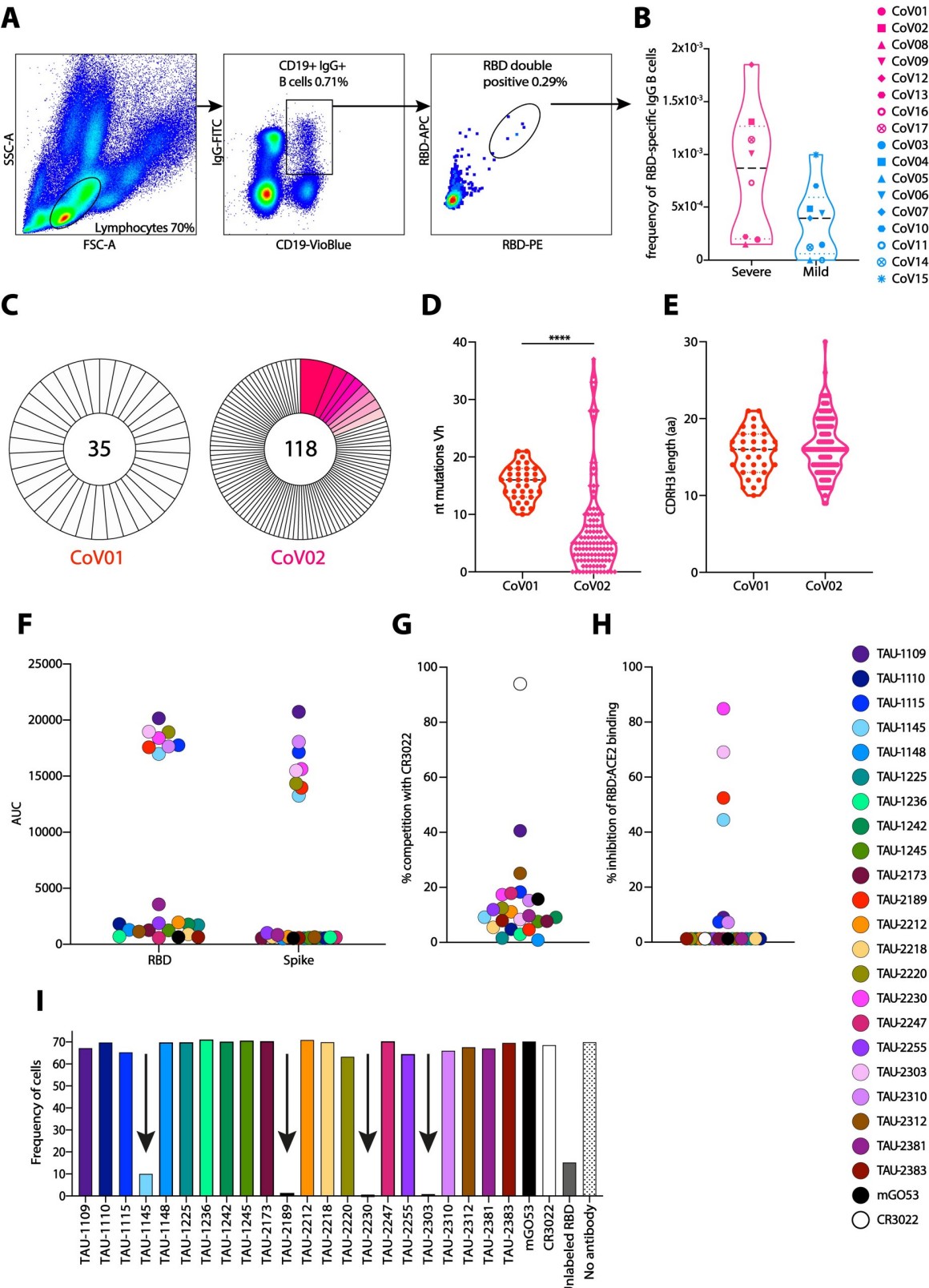

**Fig 3. Monoclonal antibodies cloned from donors CoV01 and CoV02.** (**A**) A representative flow cytometry gating strategy for staining SARS-CoV-2 RBD-specific memory B cells. (**B**) The frequencies of anti-SARS-CoV-2 RBD-specific memory B cells in COVID-19 donors

CoV01-CoV17. No PBMCs were obtained for donor CoV18, therefore this donor was not included. Symbol code is given on the right. (**C**) Pie charts representing the total number of RBD-specific memory B cell sequences (heavy chains) obtained from donors CoV01 and CoV02. The numbers in the middle of the pies represent the total number of sequences, and the pink shaded slices represent B cell clonal families. The white slices represent single sequences that were not clonal. (**D**) Number of nucleotide substitutions in VH of the B cell sequences of CoV01 (red) and CoV02 (magenta). Statistical analysis was calculated using one-way ANOVA. (**E**) Amino acid length of CDRH3 from B cell sequences isolated from CoV01 (red) and CoV02 (magenta). (**F**) AUC in ELISA for each one of the 22 cloned mAbs against SARS-CoV-2 RBD (left) and Spike (right) proteins. Color code is given on the right of (**H**). (**G**) and (**H**) Degree of mAbs competition with mAb CR3022, and ACE2, as assessed by ELISA. mGO53 was used as isotype control. (**I**) Frequency of RBD-PE stained hACE2-expressing cells identified by flow cytometry in the presence of the 22 mAbs isolated from CoV01 and CoV02. mAbs that reduced RBD binding to hACE2 are marked with black arrows. Unlabeled RBD was used as a positive control.

outside the predicted epitope, as control. We then tested the binding of other RBD-binding mAbs, as well as ACE2, to the generated RBD mutants. Mutation of a single residue G446A$_{RBD}$, that was in the core of the predicted TAU-2230 epitope, abolished completely the binding of TAU-2230, as well as the binding of other ACE2bs nAbs, TAU-2189 and 1145 (Fig 5C and 5D). Mutating residues T500D$_{RBD}$ and G496A$_{RBD}$ reduced the binding of ACE2, but did not affect TAU-2230, suggesting that TAU-2230 binding to these residues might be through interaction with the backbone atoms rather than the sidechains (Fig 5C). Antibody TAU-1109 that binds the RBD strongly, and binds outside the ACE2bs (does not compete with ACE2 for binding) served as a positive control indicating that the conformation of the mutated RBD proteins was intact. Based on these computational and mutagenesis analyses we conclude that residue G446 is central in the recognition of RBD by ACE2bs nAbs.

## Combinations of nAbs directed against different sites effectively block virus spread

Combination therapies have now become the paradigm for antiviral therapy and to avoid viral escape mutants [2,34], and were proven effective in preventing SARS-CoV-2 escape mutations [17]. We therefore investigated whether combinations of our nAbs could synergistically block viral infection and viral spread. The nAbs we isolated are directed against at least three different sites on the virus; CR3022-proximal (TAU-1109), an unknown conformational epitope (TAU-2212) and the ACE2bs (TAU-2230, -2189, -1145, and -2303). Therefore, we combined nAbs TAU-2212 and TAU-1109 at a concentration of 1 and 10 μg/mL, with each one of the anti-ACE2bs nAbs, in a double and triple mixes (Fig 6). In general, nAb TAU-2212 exhibited a more substantial effect compared to nAb TAU-1109. Also, the combination of 1 μg/ml of three nAbs (TAU-1109 + TAU-2212 + ACE2bs nAb) was in all cases more potent than 10 μg/ml of a single ACE2bs nAb (with the exception of TAU-2230 where 10 μg/ml of the single nAb was as potent as 1 μg/ml of three nAbs, Fig 6A). When using 10 μg/ml of TAU-1109 and TAU-2212 in a triple combination with ACE2bs nAbs, complete viral inhibition was achieved in all cases, although most of the effect was contributed by TAU-2212 and not TAU-1109 (Fig 6B). We conclude that combining TAU-2212 with either ACE2bs nAbs or with TAU-1109 is a potent formula for authentic SARS-CoV-2 inhibition.

## The precursors of nAbs are frequent in the BCR repertoire

The nAbs described in this study were elicited naturally following infection and therefore represent a desired response following vaccination. We therefore examined the frequency of these nAbs in the mature and naïve BCR repertoires. We were able to identify the VHJH combinations that give rise to these nAbs in 7 COVID-19 unexposed mature BCR repertoires, as well as in 42 COVID-19 unexposed naïve BCR repertoires. Moreover, the VHJH combinations for

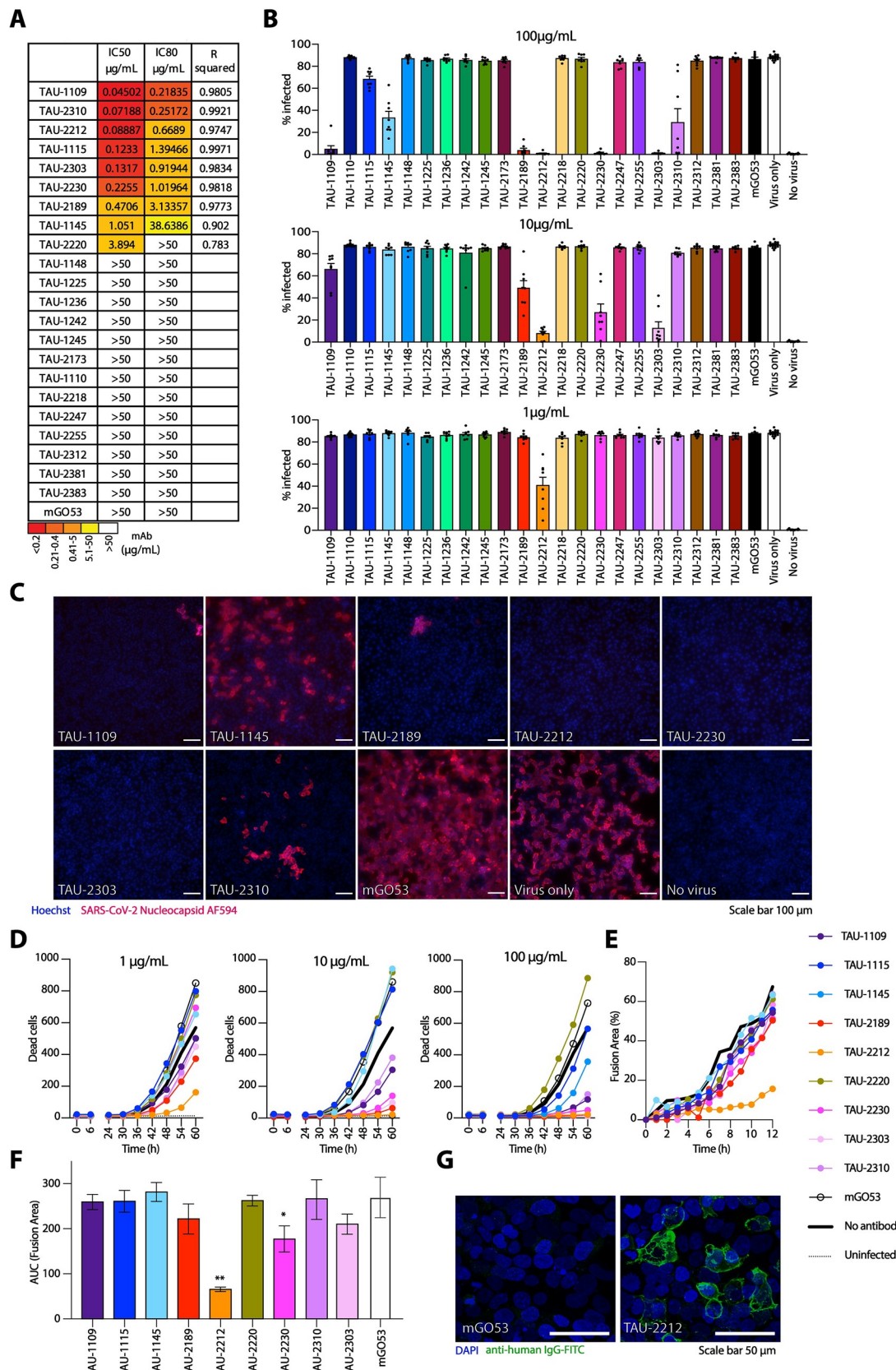

**A**

| | IC50 µg/mL | IC80 µg/mL | R squared |
|---|---|---|---|
| TAU-1109 | 0.04502 | 0.21835 | 0.9805 |
| TAU-2310 | 0.07188 | 0.25172 | 0.9921 |
| TAU-2212 | 0.08887 | 0.6689 | 0.9747 |
| TAU-1115 | 0.1233 | 1.39466 | 0.9971 |
| TAU-2303 | 0.1317 | 0.91944 | 0.9834 |
| TAU-2230 | 0.2255 | 1.01964 | 0.9818 |
| TAU-2189 | 0.4706 | 3.13357 | 0.9773 |
| TAU-1145 | 1.051 | 38.6386 | 0.902 |
| TAU-2220 | 3.894 | >50 | 0.783 |
| TAU-1148 | >50 | >50 | |
| TAU-1225 | >50 | >50 | |
| TAU-1236 | >50 | >50 | |
| TAU-1242 | >50 | >50 | |
| TAU-1245 | >50 | >50 | |
| TAU-2173 | >50 | >50 | |
| TAU-1110 | >50 | >50 | |
| TAU-2218 | >50 | >50 | |
| TAU-2247 | >50 | >50 | |
| TAU-2255 | >50 | >50 | |
| TAU-2312 | >50 | >50 | |
| TAU-2381 | >50 | >50 | |
| TAU-2383 | >50 | >50 | |
| mGO53 | >50 | >50 | |

mAb (µg/mL): <0.2 | 0.21-0.4 | 0.41-5 | 5.1-50 | >50

**B** 100µg/mL, 10µg/mL, 1µg/mL

**C** Hoechst  SARS-CoV-2 Nucleocapsid AF594    Scale bar 100 µm

**D** 1 µg/mL, 10 µg/mL, 100 µg/mL

**E**

**F**

**G** DAPI  anti-human IgG-FITC    Scale bar 50 µm

Legend: TAU-1109, TAU-1115, TAU-1145, TAU-2189, TAU-2212, TAU-2220, TAU-2230, TAU-2303, TAU-2310, mGO53, No antibody, Uninfected

**Fig 4. Neutralization of SARS-CoV-2 by mAbs.** (**A**) IC50, IC80, and R squared values for neutralization of pseudo-typed GFP-reporter viral particles. Color code is given below. (**B**) Inhibition of authentic SARS-CoV-2 infection by the 22 mAbs. Virus infected Vero E6 cells were identified by using anti-nucleocapsid antibody (GeneTex GTX135357) conjugated to AF594 after cell fixation and permeabilization. Infected cells were quantified using Image J. (**C**) Representative images of cells infected with authentic SARS-CoV-2 in the presence of mAbs. mGO53 is shown as a human isotype mAb control [42]. (**D**) Cell death induction by SARS-CoV-2 infection. Confluent Vero E6 cells were infected with SARS-CoV-2 at MOI:1 in the presence of mAbs in three concentrations as indicated. Viability of cells in five fields of view was monitored with propidium iodide every 6 h for 60 h using an Incucyte S3. (**E**) Syncytia formation inhibition as measured by the percentage of fusion area between HEK-293 target cells stably expressing hACE2 and HEK-293T cells transiently expressing SARS-CoV-2-Spike (MN908947.3) in the presence of 10 μg/mL of the appropriate antibody. The cells were incubated for 2 h, prior to imaging by the IncuCyte ZOOM system (Essen BioScience) at hourly intervals over 12 h. The percentage area of syncytia was calculated by subtracting the single cell area from the total field area using the following equation: $\%A_{syn} = \frac{A_{total} - A_{single}}{A_{total}} \times 100$. (**F**) AUC quantification of (**E**). (**G**) Confocal microscopy image of HEK-293T cells transiently expressing SARS-CoV-2-Spike (MN908947.3) and stained with TAU-2212, followed by incubation with FITC-conjugated anti-human secondary antibody. mGO53 serves as an isotype control [42].

nAbs TAU-1145, -2212, -2230, -2189, and -2310 were present within the 30 most frequent VHJH combinations. Our data indicates that SARS-CoV-2 neutralization can be achieved in many ways and that the germline gene combinations that give rise to anti-SARS-CoV-2 nAbs are relatively frequent in the healthy BCR repertoire. Therefore, given enough antigenic stimulation (which is the case during severe COVID-19), these nAbs may be produced by the majority of individuals (Fig 7).

## Discussion

In the present study we analyzed the B cell responses in convalescent COVID-19 donors, while focusing on antibodies produced in two donors with severe disease manifestations. By now, several studies showed that COVID-19 severity correlates with neutralization [7,11,35], and that anti-RBD antibody response dominates SARS-CoV-2 neutralization, although neutralizing antibodies can be directed against other viral determinants besides the RBD [9,18]. In our cohort of 18 donors all patients with symptomatic SARS-CoV-2 infection exhibited high titers of anti-RBD antibodies, in contrast to donors with milder symptoms, who exhibited a wide range of responses, ranging from high to non-detectable.

In agreement with other studies, the antibodies sequenced from memory B cells of the donors we investigated demonstrated a relatively low degree of somatic hyper mutations and clonal expansion [7,8,10,18]. Within the convalescent donors, Severe donors had higher level of shared B cell clones compared to Mild donors, likely due to increased antigenic stimulation and/or prolonged viremia to drive expansion of SARS-CoV-2-specific B cells. Moreover, the Severe donors had an increased representation of VH3-53 that was previously described as dominant in responses against SARS-CoV-2 [19,31,36]. However, the nAbs we isolated from CoV01 and CoV02 donors did not originate from this VHs and therefore constitute new VH genes that can give rise to potent anti-SARS-CoV-2 responses. Similarly to what has been reported [7,18,22], both donors, CoV01 and CoV02, exhibited neutralizing activity that was multi-clonal, *i.e.*, comprised of several neutralizing clones targeting different sites on the Spike. The contact residues of the anti-ACE2bs nAb TAU-2230 were predicted to partially overlap with ACE2bs, using similar tools to the ones we used to map the epitope of the SARS mAb 80R in 2006 [37]. This coincides with previous reports highlighting the ACE2bs as a major neutralizing determinant on the different Coronaviruses that is targeted by antibodies elicited during infection in humans [21]. Additional two groups of neutralizing determinants, outside the ACE2bs, are targeted by nAbs TAU-1109 and TAU-2212. These nAbs bind outside the ACE2bs and their epitopes, that are still to be determined, may constitute new sites of vulnerability of SARS-CoV-2, as well as targets for vaccines.

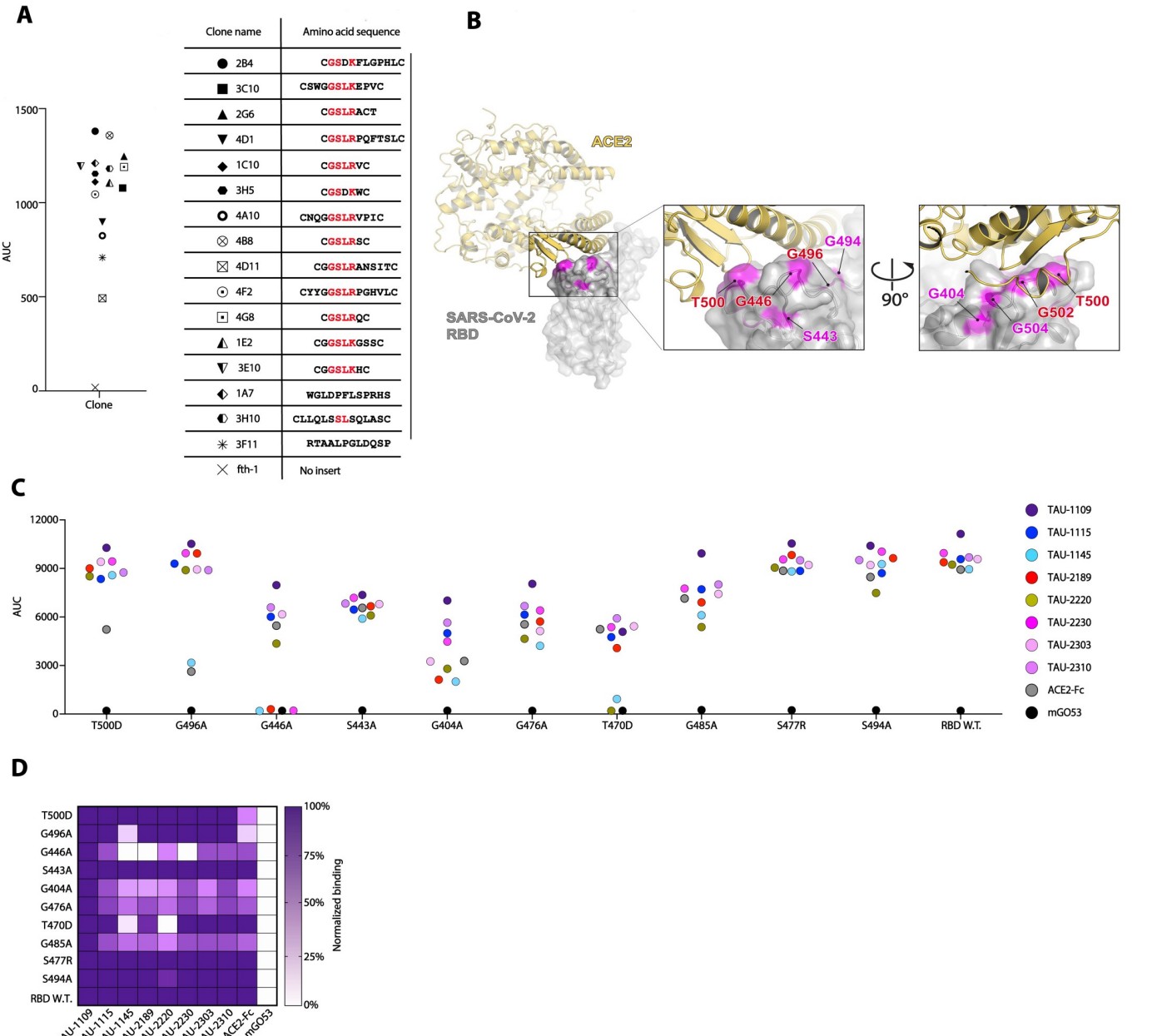

**Fig 5. TAU-2230 epitope prediction using phage-displayed affinity purified peptides and RBD mutagenesis.** (**A**) Left panel: AUC binding in ELISA of the 16 affinity-selected phage-displayed peptides to mAb TAU-2230. Fth-1 serves as a negative control (a phage that does not display an insert peptide). The symbols of the peptides are shown in the right panel. Right panel: amino acid sequences of 16 phage-displayed affinity purified peptides. Common GSLR/K motif is marked in red. The peptides were used as input for the Mapitope algorithm [32]. (**B**) Predicted TAU-2230 epitope. Left panel: ACE2:RBD complex, PDB ID 6M0J [4]. SARS-CoV-2 RBD is depicted in gray surface representation, while ACE2 is in yellow cartoon representation. The TAU-2230 predicted contact residues are shown in magenta. Middle and right panels: magnified view of the ACE2:RBD interface from two angles, the predicted TAU-2230 epitope is shown in magenta on the surface of the SARS-CoV-2 RBD. The residues that are both ACE2 and predicted TAU-2230 contacts are in red. (**C**) The effect of RBD point mutations on ACE2bs nAbs binding in ELISA (AUC, y axis). The amino acid substitutions that were introduced in SARS-CoV-2 RBD are indicated on the bottom. RBD wild-type is shown. The color-code for the different nAbs is shown on the right. (**D**) Heatmap summarizing antibody binding to mutated RBD, as well as wild-type RBD. Values for each construct are normalized to mAb TAU-1109.

The nAbs we isolated did not require substantial affinity maturation, as evidenced from the low numbers of somatic hypermutations. Moreover, bioinformatic analysis of 7 + 42 unin-fected BCR repertoires revealed that the precursors that give rise to these nAbs are abundant

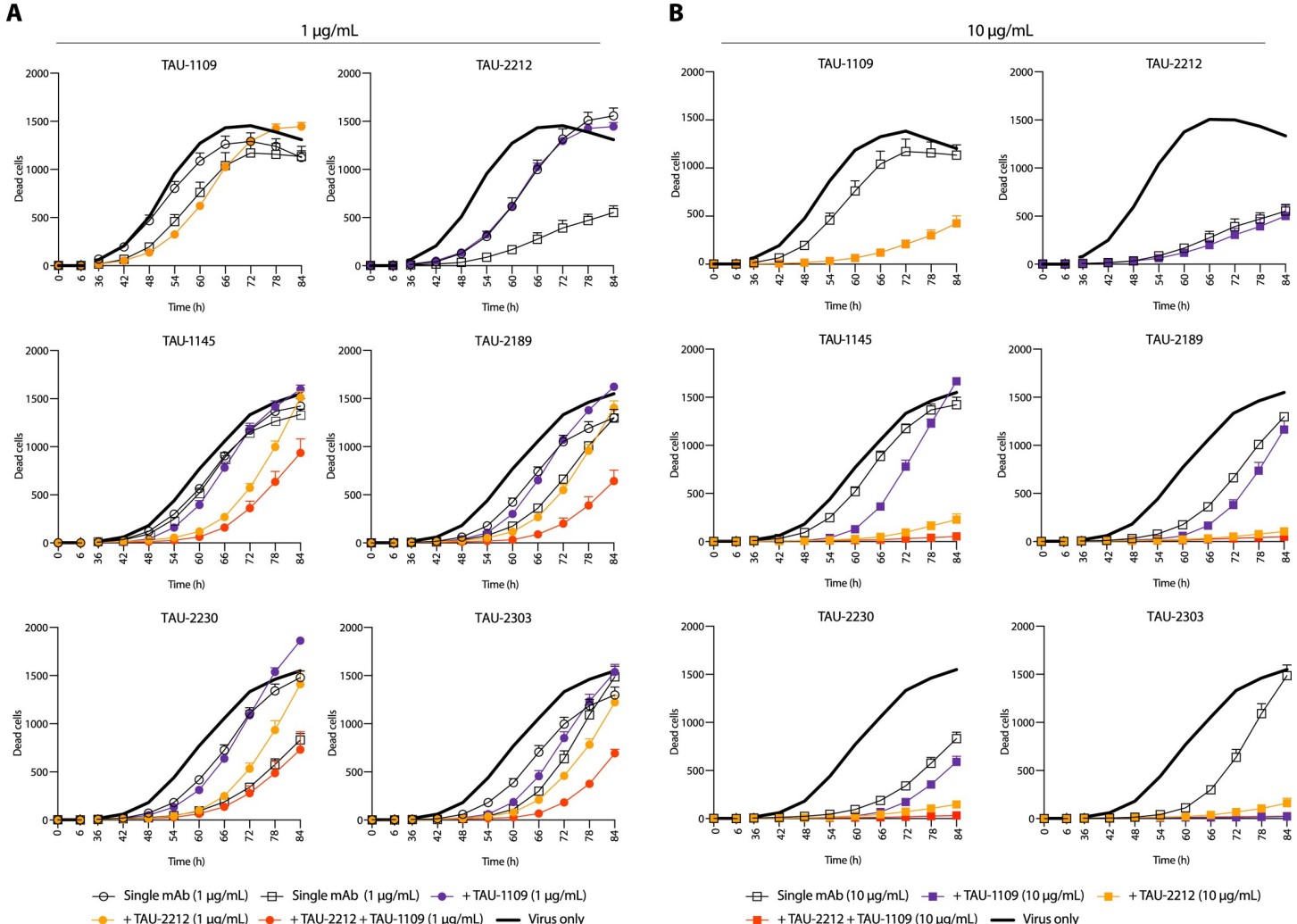

**Fig 6. Combination of nAbs targeting different sites are more effective in SARS-CoV-2 neutralization.** Virus-induced cell death traced for 84 h when nAbs were combined with the virus 1 h prior to addition to confluent Vero E6 cells at MOI:1. The antibody concentration for each nAb is 1 μg/ml, and is depicted by circles (**A**), or 10 μg/ml and is depicted by squares (**B**). In every panel, the six nAbs are indicated on the top of each graph. In each graph, empty circles or squares correspond to the effect when the indicated nAb was used alone. Full circles or squares correspond to the effect when the indicated nAb was used in combination with TAU-1109 (depicted in purple), with TAU-2212 (depicted in orange), or both (depicted in red). The black bold line shows a "virus only" (no nAb) control.

in the naïve population. Somewhat counterintuitively, many of the mild donors in our cohort did not develop anti-RBD IgGs that were able to inhibit RBD:ACE2 interaction (CoV05, -06, -07, -11, -14, and -18). This suggests that while most individuals have SARS-CoV-2 nAb precursors in their BCR repertoires, a single dose vaccine may not be sufficient to elicit protective anti-SARS-CoV-2 immunity.

Combinations of nAbs were previously shown to effectively prevent the selection of HIV-1 escape mutants [38,39]. While the mutation rates of SARS-CoV-2 are significantly lower compared to those of HIV-1, SARS-CoV-2 has been reported to mutate in culture in the presence of nAbs [40], emphasizing the importance of combination of antibodies targeting different weak spots on the virus. Recently, a combination of two nAbs targeting non-overlapping sites on SARS-CoV-2 Spike, effectively blocked the spread of SARS-CoV-2 in culture, as well as in a humanized mice model [17,41]. The combinations examined in our study can be further tested in clinical settings as a useful means of prevention and therapy of COVID-19.

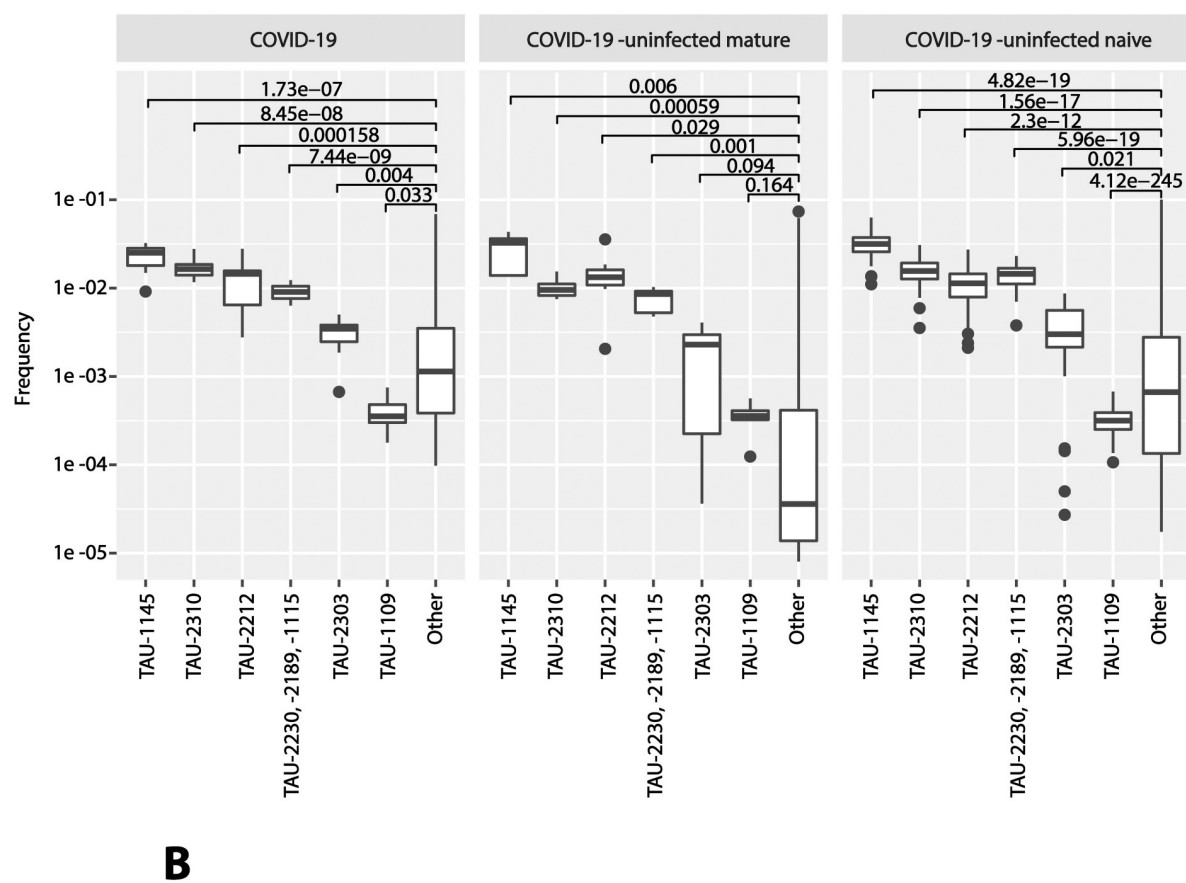

**Fig 7. VHJH precursors for anti-SARS-CoV-2 nAbs are frequent within COVID-19 infected and uninfected BCR repertoires.** (**A**) The VJ gene frequencies of anti-SARS-CoV-2 antibodies was calculated and compared to the frequency of all other VJ genes. Frequencies are shown in COVID-19 patients, in healthy matured, as well as in healthy naïve B cells. (**B**) The table lists the individual frequencies of every nAb IGHV/JH combination in each one of the three BCR datasets (COVID-19 infected, n = 13, COVID-19 un-infected, mature BCR, n = 7, COVID-19 un-infected naïve BCR, n = 42).

## Materials and methods

### Ethics statement

All COVID-19 convalescent donors, between the ages 25 and 65 recruited for this study provided a written informed consent prior to participating in this study. All donors tested positive

in at least two separate RT-PCR assays for SARS-CoV-2 infection. A sample of 150 mL whole blood was collected 1.5 months post SARS-CoV-2 infection, after two consecutive negative RT-PCR results. Tel Aviv University Institutional Review Board (IRB) approved all studies involving patient enrollment, sample collection, and clinical follow-up (protocol number 0001255–1). Donors CoV01, -02, -03, -04, -05, -06, -07, 08, -12, -13, and -18 were followed and recruited by Ichilov Tel Aviv Sourasky Medical Center (Helsinki approval number 0190-20-TLV), while donors CoV09, -10, -11, -14, -15, -16, and -17 were followed and recruited by Kaplan Medical Center in Rehovot (protocol number 0049-20-KMC).

## Cloning, expression, and purification of SARS-CoV-2 RBD and the extra-cellular human ACE2 for ELISA and FACS

Codon optimized sequences encoding the SARS-CoV-2 RBD and hACE2 were downloaded from the NCBI data base, synthesized by Syntezza-Israel, and cloned into the pcDNA 3.1 mammalian expression vector. Each construct contained an N-terminal growth factor receptor signal peptide [MKAPAVLAPGILVLLFTLVQRSNG] and two C-terminal tags—a hexa-histidine tag (HHHHHH, "His-tag") for downstream protein purification and a site-specific biotinylation tag (GLNDIFEAQKIEWHE, AviTag). Each construct was individually used to transiently transfect Expi293F cells (Thermo Fisher Scientific Inc.) using the ExpiFectamine 293 Transfection Kit (Thermo Fisher Scientific Inc.). Seven days post transfection, the cell supernatant was collected, filtered (0.22 μm), and incubated with $Ni^{2+}$-NTA agarose beads (GE Life Sciences) for 2 h at room temperature (RT). Proteins were eluted by 250 mM imidazole, buffer-exchanged to PBS x1, and then aliquoted and stored at -80˚C. The RBD protein was biotinylated with the BirA biotin-protein ligase kit (Avidity LLC, Colorado, USA) using the manufacturer's protocol.

## Isolation of plasma and PBMCs from whole blood samples

Plasma was isolated from 10 mL of each sample and the rest was used for isolation of peripheral blood mononuclear cells (PBMCs). For plasma separation, the sample was left in the collection tube (BD Vacutainer blood collection tubes, Becton, Dickinson and Company) in an upright position for 2 h followed by centrifugation at 1500 $g$ for 15 min. The upper layer containing the plasma was collected, aliquoted, and stored at -20˚C. PBMCs were isolated by diluting whole blood sample 3-fold with RPMI 1640 medium (Biological Industries Israel Beit-Haemek) and loading the material on Ficoll-Paque PLUS (GE Life Sciences) at a ratio of 1.8: 1 for phase separation according to the manufacturer's protocol. The lymphocyte containing buffy coat layer was transferred to a new vial, washed, resuspended in fetal bovine serum (FBS) containing 10% DMSO, and stored in liquid nitrogen.

## Plasma and mAb ELISA

For plasma ELISA, high-binding 96 well ELISA plates (Corning #9018) were coated with 1 μg/mL RBD in PBS x1 overnight at 4˚C. The following day, the coating was discarded, the wells were washed with "washing buffer" containing PBS x1 0.05% Tween 20, and the plates were blocked for 2 h at RT with 200 μL of "blocking buffer" containing PBS x1 3% BSA (MP Biomedicals) 20mM EDTA and 0.05% Tween20 (Sigma). Plasma samples were diluted 2-fold in blocking buffer starting from 1:100 with seven consecutive dilutions, and incubated for 1 h at RT. Plates were then washed 3 times with washing buffer before adding secondary anti-IgG, or -IgM (Jackson ImmunoResearch 109-035-088 & 109-035-129) or anti-IgA (abcam ab97215) antibodies conjugated to horseradish peroxidase (HRP) diluted 1:5000 in blocking buffer, and incubated for 45 min at RT. Following four additional washes, 100 μL of TMB/E (Millipore)

was added to each well and the absorbance at 650 nm was read after 20 min (BioTek 800 TS). Positive and negative controls were added to each plate. For mAb ELISA, high binding 96 well ELISA plates were coated with 1 µg/mL RBD, washed and blocked as described above. Antibodies were added at starting concentration of 10 µg/mL with nine additional 4-fold dilutions in PBS x1, and incubated for 1 hour at RT. Plates were then washed 3 times with washing buffer before adding secondary anti-IgG HRP-conjugated as described above. After four additional washes, 100 µL of TMB/E (Millipore) was added to each well and the absorbance at 650 nm was read after 20 min. For ACE2:RBD inhibition ELISA, high-binding 96 well plates were coated with 2 µg/mL human ACE2 in PBS x1 overnight at 4˚C. The next day, plates were washed and blocked with blocking buffer for 2 h at RT. Concurrently, biotinylated RBD was mixed with the 4-fold serial dilutions of plasma or mAbs. The RBD-plasma/mAb mix was then applied to the ACE2 coated plates and incubated for 30 min. Biotinylated RBD was detected *via* streptavidin conjugated to HRP (Jackson ImmunoResearch 016-030-084).

For CR3022 competition, ELISA plates were coated with 0.5 µg/mL RBD as described, and then washed and blocked. Serial dilutions of each mAb were added and the plates were incubated for 30 min. Next, biotinylated CR3022 at a concentration of 0.1 µg/mL was added to each well and incubated for an additional 15 min. The plates were then washed and incubated with streptavidin-HRP for 45 min. The plates were then washed and the signal detected as described above.

ELISA to validate affinity-selected phages binding to TAU-2230 mAb was carried out as follows: High-binding 96 well ELISA plates (Corning #3590) were coated with 2 µg/mL mAb TAU-2230 overnight at 4˚C. The following day, the wells were blocked for 1 h at RT with blocking solution (5% skim milk in TBS x1). After washing once with TBS x1 0.05% Tween 20, purified phages diluted in blocking solution were applied to each well, starting from $4 \times 10^8$ phages per well to give a total of five consecutive 3-fold dilutions. The plates were incubated for 1 h at RT, then washed 3 times with wash buffer, and incubated with rabbit anti-m13 polyclonal sera at a dilution of 1:5000 in blocking solution for 1 h at RT. After 3 washes, the plates were incubated at RT with a secondary anti-rabbit HRP-conjugated antibody (Jackson, West Grove, PA) at a dilution of 1:2500 in blocking solution for 45 min. The wells were then washed 3 times wash buffer and once with TBS x1, followed by the addition of the TMB/E ELISA substrate (Merck Millipore, Billerica, MA). Absorbance was measured as described.

## scRNA-seq library construction and BCR-seq data analysis

B cells were isolated from frozen PBMCs by using anti-CD19 immunomagnetic beads according to the manufacturer's protocol (CD19 Microbeads, human, Miltenyi Biotec). Purified B cells were eluted in PBS x1 containing 0.5% v/v fetal bovine serum (FBS) and 2 mM EDTA. Cells were centrifuged for 10 min at 300 x *g* and resuspended in PBS x1 containing 0.04% w/v BSA and counted. Up to 16,600 B cells were loaded on a 10x Chromium Next GEM Chip G Kit and library synthesized according to the 10x Chromium Next GEM Single Cell V(D)J Reagent Kits v1.1 according to the manufacturer's protocol (10x Genomics). BCR CDR3 sequences were enriched using the human V(D)J B cell enrichment kit (10x Genomics). Libraries were sequenced by Genewiz on a NovaSeq 6000 System using the S4 2x 150 kit (Illumina). Raw read alignment and contig assembly were performed using Cellranger V(D)J (10x Genomics, version 3.1.0), with the default compatible human reference (refdata-cellranger-vdj-GRCh38-alts-ensembl-3.1.0). Contigs were aligned to human IG loci using igBLAST-1.16.0 [43], and clones were assigned based on non-exact matching of the heavy chain sequence, using the Immcantation framework. Briefly, the normalized Hamming distance from each cell to its nearest neighbor was calculated using the Shazam R package (version

1.0.0). Based on the distribution of these distances, the cutoff to define two contigs as belonging to a single clone was set to 0.11. Clones were then assigned using the DefineClones command from the Change-O toolkit (version 1.0.0 2020.05.06 of the algorithm described by Gupta et al. [44]).

### Flow cytometry single B cell sorting and sequencing

PBMCs were rapidly thawed at 37˚C and washed in 50 mL RPMI 1640 medium, followed by B cell enrichment using anti-CD19 magnetic beads (Miltenyi Biotec, 130-050-301) according to the manufacturer's protocol. The enriched B cells were resuspended in 1% BSA in PBS x1 and 2 mM EDTA and stained with anti-CD19-FITC (Miltenyi Biotec, 130-113-645), anti-IgG-APC (Miltenyi Biotec, 130-119-772), anti-CD27-Brilliant Violet 605 (BioLegend, 302830), anti-CD38-PerCP/Cyanine5.5 (BioLegend, 356614), followed by labeled RBD via streptavidin-PE (Miltenyi Biotec, 130-106-790) and streptavidin-APC (Miltenyi Biotec, 130-106-792). Single $CD19^+IgG^+RBD^+$ cells were sorted into a 96 well plate containing 4 μL lysis buffer (12 units of RNasin Ribonuclease Inhibitor (Promega, N2511), 10 mM DTT and PBS x0.5) per well using a FACSAria III sorter (Becton Dickinson). The plates were stored in dry ice before proceeding to RNA reverse transcription and PCR amplification. For this purpose, the lysed cells were thawed on ice and the Ig genes were amplified by RT-PCR as previously described [26,27,45]. Briefly, cDNA was synthesized using random hexamer primers (Invitrogen, 48190011), and SuperScript III Reverse Transcriptase (Invitrogen, 18080085). Antibody sequences corresponding to the gamma, kappa, and lambda chains were amplified by nested PCR as previously described [27,45]. The first round of PCR ran at 94˚C for 7 min, 50 cycles of 94˚C for 30 s, 52˚C (Gamma) 50˚C (Kappa) or 58˚C (Lambda) for 20 s, and 72˚C for 40 s. The second round of PCR ran at similar conditions but the number of cycles was reduced to 40 and the annealing temperature of the gamma and lambda chains was slightly raised to 56˚C and 60˚C respectively. Second round PCR products were purified, sequenced, and annotated with IgBLAST [43]. First round PCR products were used as a template for additional amplification with specific 5' V and 3' J primers containing restriction sites for subsequent cloning into Ig expression vectors. PCR products were purified (MACHEREY-NAGEL, 740609.250), digested with the appropriate restriction enzymes (New England Biolabs), and ligated (New England Biolabs, M0202L) into the corresponding expression vectors for IgG, IgK, or IgL.

### Antibody production

Cloned mAb vectors for IgG1 heavy chain and Kappa or Lambda light chains were co-transfected at a ratio of 1:3 (H:K/L) into Expi293F cells (Thermo Fisher Scientific Inc.) using the ExpiFectamine 293 Transfection Kit (Thermo Fisher Scientific Inc.). Seven days post transfection, the cell supernatant was collected, filtered (0.22 μm) and incubated with protein A coated agarose beads (GE Life Sciences, 17519901) for 2 h at RT. The beads were then loaded onto chromatography columns, washed, and eluted using 50 mM sodium phosphate (pH 3.0) into 1 M Tris-HCl (pH 8.0). Antibodies were buffer exchanged to PBS x1, aliquoted, and stored at -80˚C.

### Antibody inhibition of SARS-CoV-2 RBD binding to hACE2-expressing cells

PE conjugated, biotinylated RBD was incubated with each mAb for 20 min at 4˚C, and then with hACE2-expressing mouse lung cells for 20 min at 4˚C. The cells were analyzed using CytoFLEX S4 (Beckman Coulter) and the PE florescence was recorded.

### Affinity selection of TAU-2230 specific peptides

Biopanning protocol was performed as previously described [33]. Briefly, approximately $1x10^{11}$ phages from the random-peptide library were mixed with 2 μg TAU-2230 mAb in TBS x1 3% BSA solution (complete to 100 μL) in 0.5 mL vials (AXYGEN, PCR tubes, PCR-05-C), and incubated for 1 h at RT. Next, 30 μL of Protein G coated magnetic beads (Invitrogen, Dynabeads Protein G) were added and the mix was incubated for additional 30 min on a rotating mixer at RT. The vials were then placed on a magnetic stand (Promega, MagneSphere Technology Magnetic Separation Stands) for 2 min to collect the beads, and the supernatant was discarded. The beads were washed 3 times with 200 μL TBS x1 0.5% Tween 20, and the bound phages were eluted twice with 105 μL of elution buffer (0.1N HCl pH 2.2, 1 mg/mL BSA) for 10 min, followed by addition of 19 μL of 1 M Tris-HCl pH 9.1. A total of three rounds of biopanning were carried out for each screen. In order to confirm binding to affinity-selected phages, monoclonal phages were produced in E. coli DH5αF$^+$, re-suspended in TBS x1 0.5% Tween 20, quantified and used in confirmatory dot blot analyses.

### Pseudo-particle preparation and neutralization assays

SARS-CoV-2-Spike pseudo-particles were obtained by co-transfection of Expi293F cells with pCMV delta R8.2, pLenti-GFP (Genecopoeia), and pcDNA 3.1 SΔC19 (ThermoFisher Scientific) at a ratio of 1:2:1, respectively, according to the manufacturer's instructions. The supernatant was harvested 72 h post transfection, centrifuged at 1500 x $g$ for 10 min to remove cell debris, and passed through a 0.45 μm filter (LIFEGENE, Israel). The pseudo-particle-containing supernatant was then concentrated to 5% of its original volume using an Amicon Ultra with a 100 KDa cutoff at 16°C (Merck Millipore).

HEK-293 cells stably expressing hACE2 were seeded into 0.1% gelatin-coated 96-well plates (Greiner) at an initial density of 0.75 x $10^5$ cells per well. The following day, concentrated pseudo-particles were incubated with serial dilutions of antibodies for 1 h at 37°C and then added to the 96 well pre-seeded plates. After 48 h, the cell medium was replaced with fresh DMEM medium excluding Phenol Red, and 24 h later, the 96-well plates were imaged by the IncuCyte ZOOM system (Essen BioScience). Cells were imaged with a 10x objective using the default IncuCyte software settings, which were used to calculate number of GFP-positive cells from four 488 nm-channel images in each well (data for each antibody concentration were collected in triplicate). The number of GFP-positive cells was normalized and converted to a neutralization percentage. IC50 and IC80 were calculated by GraphPad Prism software fitting to a non-linear regression model.

### SARS-CoV-2 infection

SARS-CoV-2 isolate USA-WA1/2020 (BEI Resources) was propagated using Vero E6 (ATCC) cells and infectious units quantified by plaque assay. For antibody neutralization assays, $10^4$ Vero E6 cells were seeded in wells of a 96-well plate and incubated overnight in DMEM 10% FBS with penicillin/streptomycin at 37°C 5% $CO_2$. The following day, titrations of mAbs were incubated with $10^4$ PFU for 1 h in DMEM 10% FBS at room temperature, then added to Vero E6 cells for up to 48 h. Infection was measured by tracking virus-induced cell death of Vero E6 cells and by intracellular staining for SARS-CoV-2 nucleocapsid, described below.

### Virus plaque assays

Virus titers were determined by plaque assays on Vero E6 monolayers. Serial dilutions of virus stocks in DMEM (Corning, #10-014-CV) were added to Vero E6 monolayers on 12-well plates

and incubated 1 h at 37˚C with rocking every 10–15 min. Virus was removed, and cells were subsequently overlaid with a 1:1 mixture of 1.2% agarose (ThermoFisher Scientific, #16500–100) and supplemented 2x MEM (2x MEM (ThermoFisher Scientific #11935046) plus 8% FBS, 2x non-essential amino acids, 2x L-glutamine, and 1.5x sodium bicarbonate). Plates were incubated at 37˚C in 5% $CO_2$ for 48 h and fixed with a mixture of 10% formaldehyde and 2% methanol (v/v in PBS) for 24 h. Agarose overlays were removed, and the monolayer was washed once with PBS x1 and stained with 0.025% Crystal Violet prepared in 2% ethanol. After 15 min, Crystal Violet was removed, and plaques were counted to determine the virus titers. Plaque assays were performed and counted by a blinded experimenter. All work with SARS-CoV-2 was conducted in Biosafety Level-3 conditions at the University of California San Diego following the guidelines approved by the Institutional Biosafety Committee.

## Immunofluorescence imaging and analysis

For viral nucleocapsid detection in Vero E6 cells by immunofluorescence, cells were washed twice with PBS x1 and fixed in 4% formaldehyde for 30 min at RT. Fixed cells were washed with PBS x1 and permeabilized for immunofluorescence using BD Cytofix/Cytoperm according to the manufacturer's protocol for fixed cells and stained for SARS-CoV-2 with a primary nucleocapsid antibody (GeneTex GTX135357) conjugated to AF594 (ThermoFisher A20185) and nuclei counterstained with Hoechst. Eight images per well were obtained using a Nikon Ti2-E microscope equipped with a Spectra III light engine (Lumencor), appropriate filter cubes (Semrock), and a Qi-2 camera. For images acquired with the Nikon Ti2-E, analysis was performed using custom-scripted code for the Fiji distribution of ImageJ (PMID 22743772) and the DeepLearning plugin StarDist as follows. Channels were separated and the Hoechst-stained nuclei were segmented using StarDist to generate individual masks for each nucleus. The AF594 channel (nucleocapsid) was selected by removing background with a rolling ball of 50 pixels. A median filter (sigma = 10) was applied to facilitate thresholding of AF594 signal. The resulting image was used to generate a mask representing all positive stain. Positive nuclei were selected by firstly eroding the StarDist-generated nuclei mask by 2 pixels to reduce potential overlap with non-positive stain. Binary reconstruction was then carried out between the resulting mask and the AF594 mask. Data from all fields of view was logged and graphed with GraphPad Prism [46].

## Virus-induced cell death

SARS-CoV-2-triggered cell death of Vero E6 cells was monitored using an Incucyte S3. Titrations of mAbs were incubated with $10^4$ PFU for 1 h in DMEM 10% FBS at RT, then added to $10^4$ confluent Vero E6 cells for up to 96 h at MOI of 1. Cell viability was monitored using 1 µg/mL propidium iodide. Five fields of view at 10x magnification representing 33.6% well coverage were monitored for changes in cell viability every 6 h. Propidium iodide-positive cells were identified using Incucyte software.

## Cell-to-cell fusion assay

HEK-293T cells were plated in a 6-well plate (Greiner) at an initial density of 0.5 x $10^6$ cells per well. In parallel, HEK-293 target cells stably expressing hACE2 were plated in a gelatin-covered 96-well plate (Greiner) at an initial density of 0.75 x $10^5$ cell per well. After 16 h incubation, the HEK-293T cells were transfected with 3 µg of pcDNA 3.1 plasmid encoding SARS-CoV-2-Spike (MN908947.3) with 19 amino acid C-terminal truncation (pSΔC19), using polyethyleneimine (PEI, Sigma Aldrich). Six hours post-transfection, the cells were washed once with complete media and collected using a cell scraper (BIOFIL, Israel). Next, 1 x $10^4$ cells were

overlaid on the target cells in the presence of 10 µg/mL of the appropriate antibody. The cells were incubated for 2 h, prior to imaging by the IncuCyte ZOOM system (Essen BioScience) at hourly intervals over 12 h. The 96-well plate (Greiner) was imaged with a 10x objective utilizing the default IncuCyte software parameters, which were used to calculate the mean confluence area of single-nuclei cells from 4 bright phase images in each well. The percentage area of syncytia was calculated by subtracting the single cell area from the total field area using the following equation:

$$\%A_{syn} = \frac{A_{total} - A_{single}}{A_{total}} \times 100$$

## Supporting information

**S1 Fig. Cloning and production of SARS-CoV-2 Spike Receptor Binding Domain (RBD) and human Angiotensin Converting Enzyme 2 (ACE2).** (**A**) Schematic illustration of the SARS-CoV-2 RBD (top) and human ACE2 (bottom) constructs that were cloned into the pcDNA 3.1 vector. Each construct contains a human secretion signal at the 5' (SP, indicated in blue) and two tags at the 3' end; a hexa histidine tag (H*6, indicated in yellow) and an AviTag ("A", indicated in red). (**B**) Representative protein gel image of the purified SARS-CoV-2 RBD (middle lane) and human ACE2 protein (right lane). The standard protein ladder is shown in the left lane.
(TIF)

**S2 Fig. Plasma responses against RBD in ELISA.** (**A-C**) IgG, IgM and IgA plasma responses of 18 SARS-CoV-2 convalescent donors (CoV01-CoV18) against the SARS-CoV-2 RBD protein. Plasma was diluted 1:100 with 7 additional consecutive 2-fold dilutions. The color-code for each donor is indicated on the right of (**B**). (**D**) Patient plasma inhibition of RBD:ACE2 binding in ELISA. Plasma was diluted 1:10 with 5 additional consecutive 4-fold dilutions. Lower OD correlates to higher inhibition. The x-axes are represented as $\log_2$ of the dilutions. The y-axis is represented as $\log_2$ of the $OD_{650}$ values.
(TIF)

**S3 Fig. IgG responses against OC43, 229E, CMV and HSV-1 in ELISA.** (**A**) Area under the curve (AUC) values for IgG binding from 18 SARS-CoV-2 convalescent donors (CoV01--CoV18) to OC43, 229E, CMV or HSV-1 (viral particles). (**B**) Correlation between anti-SARS--CoV-2 IgG serum reactivity (AUC, Fig 1A) to IgG AUC against OC43, 229E, CMV or HSV-1. AUC and correlations were calculated using GraphPad Prism software. Statistical analysis was performed using one-way ANOVA test.
(TIF)

**S4 Fig. Frequency of light chains in COVID-19 convalescent donors.** (**a**) Frequencies of the top 25 VLJL combinations in BCRs from 13 COVID-19 donors *versus* BCRs from 42 healthy individuals naïve B cells (orange bars). $-\text{Log}_{10}$ *p* values were calculated using Mann Whitney test with FDR correction. Over-represented VHJH combinations are marked in green, while under-represented are marked in red. *p* values are listed by the corresponding horizontal column. (**b**) Comparison between the frequencies of 82 VL genes in Severe *versus* Mild COVID-19 donors. Magenta right pointing horizontal bars indicate VL genes that are over-represented in Severe donors over Mild, while light blue left pointing horizontal bars indicate VL genes that are over-represented in Mild donors over Severe. Blue dashed lines correspond to $p < 0.05$ and red dashed lines corresponds to $p < 0.01$. *p* values were calculated using Mann Whitney test. (**c**) Volcano plot showing *p* value as calculated in (**b**) and normalized fold counts

for every segment. $^{**}$ $p < 0.01$ $^{***}$ $p < 0.001$ $^{****}$ $p < 0.0001$.
(TIF)

**S5 Fig. RBD-positive memory B cells in donors CoV01-17.** Flow cytometry plots showing APC-RBD and PE-RBD double stained memory B cells for CoV01-17 (no PBMCs were obtained for CoV18). The frequencies of double positive B cells are indicated within each plot.
(TIF)

**S6 Fig. Activity of anti-SARS-CoV-2 mAbs in ELISA.** (**A**) and (**B**) CoV01 and CoV02 mAbs, respectively, binding to SARS-CoV-2 RBD (left) and Spike trimer (right). The color-code is indicated to the right of each graph. (**C**) Antibody inhibition of RBD:ACE2 binding in ELISA. Antibodies were assayed at 300 nM with 6 additional consecutive 4-fold dilutions. The y-axis is represented as $\log_2$ of the $OD_{650}$ values. Lower OD indicates higher mAb inhibition. (**D**) Antibody competition with biotinylated-CR3022. Lower $OD_{650}$ values indicate a higher level of competition between the mAbs.
(TIF)

**S7 Fig. Affinity measurements of anti-SARS-CoV-2 mAbs.** (**A**) SPR sensograms showing binding of injected SARS-CoV-2 RBD at six different concentrations (15.6 nM, 31.25 nM, 62.5 nM, 125 nM, 250 nM and 500 nM) to immobilized anti-SARS-CoV-2 TAU mAbs (0.5 μg/ml). mGO53 was used as isotype control. SPR assays were performed on a Biacore T200 instrument at 25°C. Three replicates were performed for each mAb and all samples were diluted in HBS-EP buffer (0.01 M HEPES, 0.15 M NaCl, 0.003 M EDTA, 0.05% Tween 20, pH 7.4). Sensograms were fitted to 1:1 binding model using non-linear regression in the biaevaluation software. $K_D$ was calculated using the ratio of the kinetic constants $K_D = K_d/K_a$. (**B**) Rmax, KD, and U-value values.
(TIF)

**S8 Fig. mAb inhibition of SARS-CoV-2 RBD binding to hACE2-expressing cells.** (**A**) Flow cytometry plots showing hACE2-expressing cells stained with RBD-PE ("no Ab", left) and unstained hACE2-expressing cells (right). (**B**) Anti-SARS-CoV-2 mAbs were pre-incubated with RBD-PE followed by incubation with hACE2-expressing cells. Unlabeled RBD was used as a positive control, and mGO53 as a negative control [42]. The frequencies of PE positive cells are indicated. (**C**) Mean fluorescence intensity (MFI) of RBD-PE stained hACE2-expressing cells identified by flow cytometry in the presence of mAbs (mAbs that reduced RBD-PE staining are marked with black arrows).
(TIF)

**S9 Fig. HEK-293 cell infection with pseudo-typed SARS-CoV-2 in the presence of anti-SARS-CoV-2 nAbs.** Representative images of HEK-293 cells stably expressing hACE2 infected with SARS-CoV-2-Spike GFP-expressing pseudo-particles in the presence of mAbs TAU-1145, -2220, -2189, -1115, -2212, -2230, -2303, -2310, and -1109, as well as the negative control mAb mGO53 [42]. Cells were imaged 24 h post infection using IncuCyte ZOOM.
(TIF)

**S10 Fig. Vero E6 cells infected with SARS-CoV-2 in the presence of anti-SARS-CoV-2 mAbs.** Images of Vero E6 cells infected with authentic SARS-CoV-2 following fixation and staining with nucleocapsid antibody AF594 and Hoechst nuclear staining. Virus and antibodies were pre-incubated for 1 h prior to infection. mGO53 is shown as a human isotype control [42].
(TIF)

**S11 Fig. Dose-dependent cell death prevention by mAbs following infection with SARS--CoV-2.** Confluent Vero E6 cells were infected with SARS-CoV-2 at MOI:1 in the presence of a titration of the 22 mAbs along with controls. Four different concentrations used are labeled black (100 μg/mL), red (10 μg/mL), blue (1 μg/mL), and yellow (0.1 μg/mL). Viability of cells in five fields of view was monitored with propidium iodide every 6 h for 60 h using an Incucyte S3.
(TIF)

**S12 Fig. TAU-2212 binding to SARS-CoV-2 Spike-expressing HEK-293 cells.** (**A**) Confocal microscopy images of HEK-293 cells transiently expressing SARS-CoV-2-Spike (MN908947.3) and stained with TAU-2212 and a mix of TAU-1109 and TAU-2230 followed by incubation with FITC-conjugated anti-human secondary antibody. mGO53 serves as an isotype control [42]. (**B**) Flow cytometry plots of Expi293F cells transiently expressing SARS--CoV-2-Spike incubated with TAU-1109, TAU-2212 and TAU-2230 and stained with APC-conjugated anti-human secondary antibody. mGO53 serves as an isotype control [42].
(TIF)

**S13 Fig. TAU-2230 predicted epitope candidates.** Three predicted epitope-candidates, Clusters A, B and C are depicted in red, magenta and green, respectively. The predicted residues are shown in spacefill on the surface of SARS-CoV-2 RBD (PDB ID 6M0J [4]). The amino acid residues comprising each cluster are listed. The prediction was conducted using the D = 9 A, ST = 6, f = 0 parameters.
(TIF)

**S1 Table. COVID-19 convalescent donors participated in this study.**
(TIF)

**S2 Table. Anti-SARS-CoV-2 monoclonal antibodies isolated and cloned in this study.** The donor from whom each mAb was isolated is indicated to the left. Neutralizing mAbs are colored in red. Clonal relatives are shaded in the same color.
(TIF)

# Acknowledgments

We thank the members of the Freund Lab for fruitful discussions and assistance. We thank Rony Dahan for the Spike protein. We thank Eran Bacharach for the hACE2-expressing cell line. We thank Noam Solomon and Luis Voloch from ImmuneAi for discussions. We thank Sackler Interdepartmental Core Facility for help with single cell sorting. We thank E. Pichinuk and the Blavatnik Center for Drug Discovery for help with SPR experiments. We thank Nikon Imaging Inc. for the loan of the Ti2 microscope system. J.M.G. acknowledges the kind support of Dr. Peter Kraus. N.T.F. acknowledges the kind support of L. Cohen. The following reagent was deposited by the Centers for Disease Control and Prevention and obtained through BEI Resources, NIAID, NIH: SARS-Related Coronavirus 2, Isolate USA-WA1/2020, Heat Inactivated, NR-52286.

# Author Contributions

**Conceptualization:** Michael Mor, David Hagin, Natalia T. Freund.

**Data curation:** Modi Safra, Elad Chomsky, Hila Sharim, Jonathan M. Gershoni, Oren Zimhony, Gur Yaari, David Hagin, Ben A. Croker, Natalia T. Freund.

**Formal analysis:** Modi Safra, Elad Chomsky, Ksenia Polonsky, Hila Sharim, Gur Yaari, Natalia T. Freund.

**Funding acquisition:** Natalia T. Freund.

**Investigation:** Michael Mor, Michal Werbner, Joel Alter, Jamie C. Lee, Smadar Hada-Neeman, Cameron J. Nowell, Alex E. Clark, Anna Roitburd-Berman, Noam Ben-Shalom, Michal Navon, Dor Rafael, Evgeny Kiner, Oren Kobiler, Aaron F. Carlin, Moshe Dessau, Meital Gal-Tanamy, Ben A. Croker, Natalia T. Freund.

**Methodology:** Michael Mor, Michal Werbner, Joel Alter, Moshe Dessau, Meital Gal-Tanamy, Ben A. Croker, Natalia T. Freund.

**Project administration:** Eric R. Griffis, Natalia T. Freund.

**Resources:** Cameron J. Nowell, Eric R. Griffis, Sandra Lawrynowicz Leibel, Aaron F. Carlin.

**Software:** Modi Safra, Elad Chomsky, Ksenia Polonsky, Cameron J. Nowell, Hila Sharim, Gur Yaari.

**Supervision:** Gur Yaari, Ben A. Croker, Natalia T. Freund.

**Visualization:** Michael Mor, Michal Werbner, Joel Alter, Modi Safra, Ksenia Polonsky, Moshe Dessau, Meital Gal-Tanamy, Ben A. Croker, Natalia T. Freund.

**Writing – original draft:** Michael Mor, Ben A. Croker, Natalia T. Freund.

**Writing – review & editing:** Michael Mor, Jonathan M. Gershoni, Oren Kobiler, Ben A. Croker, Natalia T. Freund.

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
