## [Decision Letter · Decision Letter 0]

15 Oct 2020

Dear Dr. Freund,

Thank you very much for submitting your manuscript "Multi-Clonal SARS-CoV-2 Neutralization by Antibodies Isolated from Severe COVID-19 Convalescent Donors" for consideration at PLOS Pathogens. As with all papers reviewed by the journal, your manuscript was reviewed by members of the editorial board and by several independent reviewers. The reviewers appreciated the attention to an important topic. Based on the reviews, we are likely to accept this manuscript for publication, providing that you modify the manuscript according to the review recommendations.

The reviewers have identified some issues that should be addressed.

Sincerely,

Kanta Subbarao

Section Editor

PLOS Pathogens

Kanta Subbarao

Section Editor

PLOS Pathogens

Kasturi Haldar

Editor-in-Chief

PLOS Pathogens

orcid.org/0000-0001-5065-158X

Michael Malim

Editor-in-Chief

PLOS Pathogens

orcid.org/0000-0002-7699-2064

The reviewers have identified some issues that should be addressed.

Reviewer Comments (if any, and for reference):

Reviewer's Responses to Questions

**Part I - Summary**

Reviewer #1: Mor and colleagues investigate the antibody response in COVID patients comparing those with mild and severe symptoms. Their findings from plasma analysis confirm recent reports that severe cases have more IgG antibodies and more virus neutralizing activity. Molecular analysis of antibody sequences identified some VHJH that were enriched in COVID infected individuals, and two sequences that were uniquely overexpressed in the severe group. Recombinant antibodies to the RBD were derived from two patients and characterized. They recognize distinct epitopes on RBD and are neutralizing.

Several groups have recently reported on the antibody response and on monoclonal antibodies to SARS-CoV-2, which diminishes the novelty of the findings. Nevertheless, the experiments and analyses are of very high quality and importance, carefully reported, and important to the field. Rigorous methodology was applied.

Reviewer #2: This is a well-written manuscript describing the isolation and characterisation of SARS-CoV-2 neutralizing antibodies from infected individuals. The authors use the RBD to select B cells. Sequence analysis of SARS-CoV-2 specific mAbs are used to analyze the clonality of the immune response and are compared to those of “non-covid” individuals. Below are some points that need to be addressed. This is a timely study and adds to the body of research on Ab responses to SARS-CoV-2 infection.

**Part II – Major Issues: Key Experiments Required for Acceptance**

Reviewer #1: Two major issues:

- Antibodies are made of heavy and light chains (H and L). Particular VHJH sequences were found to be enriched in COVID patients. The same analysis should be performed for the light chains, and if possible for the heavy and light chains combined.

- It is stated that ‘data are available with no restrictions’ but could not find table with all antibody sequences related to Figure 2, and the full sequence of the cloned monoclonal antibodies is missing. This information should be included.

Reviewer #2: (No Response)

**Part III – Minor Issues: Editorial and Data Presentation Modifications**

Reviewer #1: - ‘live SARS-CoV-2’: presumably the authors mean ‘authentic SARS-CoV-2’ as opposed to pseudo virus, consider re-wording

- first part of the introduction could be improved

- third paragraph of the results: the main text should explain how the RBD positive cells were identified

- Figure 7 is hard to interpret

- Discussion: “Severe donors had higher level of shared B cell closed(?) compared to…”

- many have shown the effects of combination therapy (e.g. Regeneron antibodies) so the following should be modified or removed: “This is the first demonstration that combinations of neutralizing anti-SARS-CoV-2 nAbs can effectively block the spread of live SARS-CoV-2”

Reviewer #2: The authors discuss pre-existing immunity to HCoV and cross-reactivity between SARS-CoV-2 in the first section of the results. However, they are using samples collected after SARS-CoV-2 infection so this section should be changed to take this into account.

The authors isolate B cells binding to the RBD bait and analyze the sequences generated. However, in the downstream analysis it appears that not all the expressed mAbs bind to S. What do the authors think is the reason for this? If they are false positives in the sorting step then these sequences should not be included in the sequence analysis. This needs to be addressed in the text.

It is not clear how the 9 and 13 mAbs from donors CoV01 and CoV02 were selected for further analysis. This should be clarified in the text.

In figure 6b and 6c, the mAb alone (1109 or 2212) at 2ug/mL or 20ug/mL should be shown on the graph to compare with the combinations of mAbs and better show the synergistic effects.

Is it not expected that the Ab response would be multi-clonal? Many other studies have shown that there are several neutralizing epitopes on S. These papers should be referenced in the discussion section.

The authors highlight that some individuals with mild COVID-19 do not produce anti-RBD-IgG that could inhibit RBD:ACE2 interaction. Do these donors have neutralizing serum? If yes, would this not suggest that they produce nAbs targeting the other neutralizing epitopes identified?

In the final paragraph of the discussion, the authors conclude that this is the first study to show combinations of mAbs can block spread of live SARS-CoV-2. However, the data for a mAb on its own is not shown for comparison. The relevance of this observation should be included within the discussion. Why is the spreading assay more relevant than a neutralization assay for example? Other labs have also tried combinations and these should be referenced/discussed.

More references are required to other studies on SARS-CoV-2 Ab responses are needed. In particular the part referring to the relationship between disease severity and magnitude of the Ab response.

Page 7: what was the potency of the live virus neutalization compared to the pseudovirus?

Page 7: TAU-2212 does not bind S in ELISA but does on the cell surface and has neutralizing activity. The authors suggest it may be binding an epitope involving more than one subunit. The authors do not state what form the S is that is used in ELISA. Is it a stabilised trimer?

Page 9: The data presented does not justify the conclusion that “these nAbs can be produced by the majority of individuals”. “may” would be more appropriate.

PLOS authors have the option to publish the peer review history of their article (what does this mean?). If published, this will include your full peer review and any attached files.

Reviewer #1: No

Reviewer #2: No
---

## [Editor Report · Decision Letter 1]

25 Nov 2020

Dear Dr. Freund,

We are pleased to inform you that your manuscript 'Multi-Clonal SARS-CoV-2 Neutralization by Antibodies Isolated from Severe COVID-19 Convalescent Donors' has been provisionally accepted for publication in PLOS Pathogens.

Best regards,

Kanta Subbarao

Section Editor

PLOS Pathogens

Kanta Subbarao

Section Editor

PLOS Pathogens

Kasturi Haldar

Editor-in-Chief

PLOS Pathogens

orcid.org/0000-0001-5065-158X

Michael Malim

Editor-in-Chief

PLOS Pathogens

orcid.org/0000-0002-7699-2064
---

## [Editor Report · Acceptance letter]

19 Jan 2021

Dear Dr. Freund,

We are delighted to inform you that your manuscript, "Multi-Clonal SARS-CoV-2 Neutralization by Antibodies Isolated from Severe COVID-19 Convalescent Donors," has been formally accepted for publication in PLOS Pathogens.

Best regards,

Kasturi Haldar

Editor-in-Chief

PLOS Pathogens

orcid.org/0000-0001-5065-158X

Michael Malim

Editor-in-Chief

PLOS Pathogens

orcid.org/0000-0002-7699-2064